# Epidemiology of Tick-Borne Encephalitis (TBE) in Germany, 2001–2018

**DOI:** 10.3390/pathogens8020042

**Published:** 2019-03-29

**Authors:** Wiebke Hellenbrand, Teresa Kreusch, Merle M. Böhmer, Christiane Wagner-Wiening, Gerhard Dobler, Ole Wichmann, Doris Altmann

**Affiliations:** 1Immunization Unit, Robert Koch Institute, Seestraße 10, 13353 Berlin, Germany; HellenbrandW@rki.de (W.H.); WichmannO@rki.de (O.W.); 2Department for Infectious Disease Epidemiology, Bavarian Health and Food Safety Authority, Veterinaerstr. 2, 85764 Oberschleissheim, Germany; Merle.Boehmer@lgl.bayern.de; 3State Health Office Baden-Wuerttemberg, Nordbahnhofstr. 135, 70191 Stuttgart, Germany; Christiane.Wagner-Wiening@rps.bwl.de; 4Bundeswehr Institute of Microbiology, Neuherbergstrasse 11, 80937 Munich, Germany; gerharddobler@bundeswehr.org; 5Infectious Disease Data Science Unit, Robert Koch Institute, Seestraße 10, 13353 Berlin, Germany; AltmannD@rki.de

**Keywords:** Tick-borne encephalitis (TBE), epidemiology, incidence, risk areas, clinical manifestations, temporospatial distribution, vaccination, Germany

## Abstract

We reviewed tick-borne encephalitis (TBE) surveillance and epidemiology in Germany, as these underlie public health recommendations, foremost vaccination. We performed descriptive analyses of notification data (2001–2018, n = 6063) according to region, demographics and clinical manifestations and calculated incidence trends using negative binomial regression. Risk areas were defined based on incidence in administrative districts. Most cases (89%) occurred in the federal states of Baden-Wurttemberg and Bavaria, where annual TBE incidence fluctuated markedly between 0.7–2.0 cases/100,000 inhabitants. A slight but significantly increasing temporal trend was observed from 2001–2018 (age-adjusted incidence rate ratio (IRR) 1.02 (95% confidence interval (CI): 1.01–1.04)), primarily driven by high case numbers in 2017–2018. Mean incidence was highest in 40–69-year-olds and in males. More males (23.7%) than females (18.0%, *p *= 0.02) had severe disease (encephalitis or myelitis), which increased with age, as did case-fatality (0.4% overall; 2.1% among ≥70-year-olds). Risk areas increased from 129 districts in 2007 to 161 in 2019. Expansion occurred mainly within existent southern endemic areas, with slower contiguous north-eastern and patchy north-western spread. Median vaccination coverage at school entry in risk areas in 2016–2017 ranged from 20%–41% in 4 states. Increasing TBE vaccine uptake is an urgent priority, particularly in high-incidence risk areas.

## 1. Introduction

In Germany, infection with tick-borne encephalitis (TBE) virus (TBEV) is transmitted mainly by the tick *Ixodes ricinus*. TBEV infection is asymptomatic or presents with non-specific symptoms, including fever and malaise, in 70%–95% of cases [1,2]. The remainder develops neurological manifestations ranging from uncomplicated meningitis to severe meningoencephalitis and/or myelitis [1,2,3,4]. TBE is vaccine-preventable, with two highly effective vaccines based on the European TBEV subtype available in Europe [5,6]. 

Serological evidence for TBE in humans and mice in Germany was first reported by Sinneker in various regions of Eastern Germany in 1959 [7], followed by serologic diagnosis of symptomatic TBE in humans 1960–1961 in Saxony [8,9]. Süss et al. [10,11] later reviewed TBE cases in Eastern Germany, reporting 1200 cases ascertained from 1960 to 1970, many occurring during outbreaks related to the consumption of raw milk. From 1960 to 1970, TBE incidence in Eastern Germany decreased from 0.7 to 0.02 cases/100,000 TBE cases/100,000 inhabitants, and only isolated TBE cases were ascertained in the 1970s and 1980s. However, after the reunification of Germany in 1990, isolated TBE cases were again diagnosed in Eastern federal states, namely in Thuringia and Saxony [10]. TBE cases in Thuringia occurred mainly in regions adjacent to existing Bavarian TBE risk areas and in Saxony adjacent to known risk areas in the Czech Republic [12]. 

In the Federal Republic of Germany, the first TBE case was described near Passau in Bavaria in 1964 [13], with further case descriptions from Baden-Wurttemberg (BW) and Bavaria (BY) in the 1960s and 1970s [13,14,15,16,17]. Ackermann described 149 and 51 TBE cases ascertained in southern Germany from 1964 to 1977 [18] and 1978–1980 [19], respectively. Testing of 8526 residual laboratory sera collected from 1980–1985 revealed neutralising TBE antibodies in 2% of sera from Bavaria and 1.1% from Baden-Wurttemberg [19]. In the 1990s an increasing number of cases were reported from the Odenwald region in northern Baden-Württemberg/southern Hesse, and sporadic cases from more western regions in the Saarland and Rhineland-Palatinate [10,20,21,22,23]. 

In the mid-1990s meetings with public health authorities were held in Germany leading to the definition of administrative districts as TBE-risk areas based on numbers of TBE cases ascertained in surveys [3,24]. In 1997 the National Immunization Technical Advisory Group (STIKO) recommended vaccination for persons with tick-exposure in these risk areas [25]. A risk area was defined as a district with at least 5 TBE cases in any 5-year period or with at least 2 cases in any one calendar year [24,26,27,28]. In 2001 TBE was made a statutorily notifiable disease according to the German Protection Against Infection Act (IfSG). From 2001 to 2005, the data base was restricted to the respective preceding 20-year period, with data for the years 1982–2001 originating from surveys and for 2002–2005 from statutory surveillance [29,30,31,32,33]. The number of districts classified as risk areas increased from 55 in 1997 to 96 in 2006. This development occurred almost exclusively within a largely contiguous area spanning the states of Bavaria, Baden-Wurttemberg, southern Hesse and southern Thuringia, with a further isolated district each in mid-northern Hesse and Rhineland-Palatinate. As of 2007, risk areas were defined using notification data only using an incidence-based approach [34]. 

Surveillance provides the basis for public health recommendations–foremost vaccination–for TBE prevention. Thus, we aim to review TBE surveillance and epidemiology in Germany from 2001 to 2018, including the definition of TBE risk areas and their distribution over time as of 2007. 

## 2. Results

### 2.1. Tick-Borne Encephalitis (TBE) Incidence, Seasonality, Demographics and Trends 

The majority of the 6,063 notified TBE cases from 2001–2018 originated in Baden-Wurttemberg (BW) and Bavaria (BY) (89.0% of cases with a reported place of infection (PoI) in Germany), followed by Hesse, Saxony, Thuringia, and Rhineland-Palatinate (Figure 1). All current risk areas but two (in Saarland and Lower Saxony) are located in these states (see below). The annual number of reported cases ranged from 195 to 583, the incidence in the two most affected states of BW and BY ranging from 0.7 to 2.0 cases/100,000 inhabitants (mean 1.2), with peaks in 2005–2006, 2011, 2013 and 2017–2018 (Figure 1 and Figure 2). Increases in incidence did not follow a uniform pattern. For instance, the increase in 2005 was more marked in BY, while the further increase in 2006 occurred almost entirely in BW and Hesse (Figure 1) with case numbers in BY decreasing slightly, although remaining markedly above average. The peaks in 2011, 2013 and 2017 were due to increases in both BW and BY, while the peak in 2018 again occurred due to a further increase in BW, despite a slight decrease in case numbers in BY, as in 2006. Of the 222 cases reported with the PoI exclusively outside of Germany, 177 (79.7%) were from 6 countries (Austria (94), Czech Republic (20), Poland (17), Switzerland (16), Sweden (16) and Italy (14)).

Of all cases, 91.1% occurred from May to October, with 49.4% occurring in June and July alone (Figure 2). Peak years were characterized by higher case numbers especially in these two months, but often additionally by extension of disease activity into late summer and early fall (2005, 2006, 2013, and 2017). In 2018 and to a lesser extent in 2011, cases occurred earlier than average. 

TBE incidence was higher in males (63.5% of all cases) than in females in all age groups (Figure 3). Incidence was highest in adults aged 40 to 69 years, peaking at a younger age in women (median age: 47 years (interquartile range (IQR): 34–58) than in men (median age 49 years (IQR: 36–62, *p* = 0.0001 (Kruskal–Wallis test), Figure 3). 

Analysis of age-specific and total TBE incidence over time in the two most affected states BY and BW showed marked annual fluctuation with a slightly increasing temporal trend from 2001 to 2018 of 2% (95% confidence interval (CI) 0.8%–3.6%) annually (Figure 4). We examined possible differences among age-specific trends by including interaction terms in our negative binomial regression model, but found no significant improvement using the likelihood ratio test. Thus, the final model assumes a uniform trend for all age groups. The seasons 2017 and 2018 had unusually high case numbers; in particular, TBE incidence was higher than in previous peak years among 50- to 69-year-olds and in 2018 among those 70 years and older (Figure 4). Therefore, we repeated these trend analyses using only data from 2001–2016 and found no significant trend (incidence rate ratio (IRR): 1.00, 95% CI: 0.97–1.03). Thus, a decrease in case numbers in the near future, as would be in keeping with the fluctuations of previous years, might lead to loss of the significantly increasing trend observed from 2001 to 2018. 

### 2.2. TBE Risk Areas

In 2007, 129 districts were classified as risk areas based on TBE cases notified from 2002–2006 (Figure 5a). By 2019, this had increased to 161 districts based on data from 2002 to 2018 (Figure 5b). Most additional risk areas border on previously existent ones in BW, BY, Hesse and Thuringia. In recent years marked northeastward expansion occurred into Saxony, where 4 districts have been classified as risk areas since 2016. In addition, one district each in Saarland and in Lower Saxony were classified as risk areas in 2010 and 2019, respectively (Figure 2B). TBE cases continued to be notified from almost all districts ever classified as TBE risk areas at least sporadically, with the exception of a marked decrease in incidence in 5 contiguous districts in northeastern BW/northwestern BY in recent years (Figure 5). In these districts, five-year incidences ranged from 2.0–8.0 TBE cases/100,000 inhabitants in 2002–2006, but decreased to 0 from 2014–2018 in all but one district, where a single case with an additional possible PoI was reported.

The median incidence in the most recent 5-year interval 2014–2018 in districts defined as risk areas was 3.7 cases/100,000 inhabitants, ranging from a minimum of 0 to a maximum of 48.0. Of the 1597 TBE cases reported with a PoI in a risk area in this time period, 1132, or 70.9%, occurred the risk areas with a TBE incidence in the top tertile (>=6.4 cases/100,000 inhabitants). Sporadic TBE cases have been notified in all federal states except the city states of Hamburg and Bremen. The dots in Figure 5a,b portray sporadic cases with reported exposure in districts not classified as TBE risk areas in 2002–2006 (n = 83) and 2014–2018 (n = 52), respectively. Over half of sporadic cases were notified from states with risk areas (53.8% in 2002–2006 and 56.6% in 2014–2018, Figure 5a,b)). In 2002–2006, the distribution of sporadic cases heralded future risk areas in BY, BW and Saxony (Figure 5a). By 2014–2018, an increasing number of sporadic cases occurred in Lower Saxony and Northrhine-Westphalia. In both periods, isolated sporadic cases occurred in three eastern federal states Brandenburg, Mecklenburg-Western Pomerania and Saxony-Anhalt, where TBE had been endemic in the 1960s and 1970s (see above). 

### 2.3. Vaccination Coverage 

Vaccination coverage in children at school entry in 2016/2017 in risk areas varied widely (Figure 6). Median coverage was highest in Bavaria and Hesse and lowest in Baden-Wurttemberg. Although median coverage was significantly higher in risk areas with a TBE incidence in the highest tertile in the 5 year period 2012–2016 (39.4%) than in risk areas with lower TBE incidences (28.1%, Figure 6), there was substantial overlap.

In the aforementioned 5 districts in northern BW and BY with a marked decrease in TBE incidence (LK Neckar-Odenwald-Kreis, LK Main-Tauber-Kreis, SK Würzburg/LK Würzburg, and LK Kitzingen), vaccination coverage at school entry ranged from 31.5% to 48.6% and was thus no higher than in many other regions with persistent occurrence of cases.

### 2.4. Clinical Aspects

Of the 1636 TBE cases reported from 2015 to 2018, 770 (47.1%) were reported with non-specific symptoms only and 866 (52.9%) with manifestations of the central nerval system (CNS) (Figure 7). CNS manifestations were less frequently reported in female (50.7%) than in male (54.4%) cases, but this was not statistically significant (age-adjusted odds ratio (OR): 0.87, 95% CI: 0.71–1.06). Meningitis was reported in 31.5% of cases (females: 30.8%; males: 32.7%) and encephalitis or myelitis (with or without meningitis) in 21.5% (females: 18.0%; males: 23.6%, *p* = 0.024). Myelitis was rare, reported only in 61 patients (3.7%), more often in males (4.5%) than females (2.6%, *p* = 0.05). The more severe clinical manifestations encephalitis or myelitis increased markedly among cases aged 60 years and older, while the proportion with isolated meningitis decreased in the older age groups (Figure 7, *p* trend < 0.0001). 

Hospitalization was reported in 87.4% of cases notified from 2015 to 2018 (1,389/1,588 cases with information available), lowest in children under 5 years of age (60.0%) and highest in patients 70 years and older (93.8%, *p* trend = 0.07). Persons reported with CNS symptoms were more likely to be hospitalized (meningitis only: 93.2%; encephalitis or myelitis: 96.0%) than those without (79.5%, *p* chi2 < 0.0001). Duration of hospitalization was available for 509 of these hospitalized cases (32.0%) and was longer in patients with encephalitis or myelitis compared to patients with meningitis only (median of 11 vs. 8 days, with 24.4% vs. 9.4% hospitalized >2 weeks (*p* < 0.0001)). Among cases with non-specific symptoms the median duration of hospitalization was also 8 days with 14.7% requiring hospitalization >2weeks.

Among the 6,035 cases notified from 2001 to 2018 with reported outcome, there were 25 deaths (case-fatality: 0.4%). Case fatality was non-significantly higher in males (0.5%) than females (0.3%, *p *= 0.38). Deaths occurred in patients aged 28 to 94 years, but 23 were in patients aged 54 years and older. Case-fatality increased with age, to 2.1% in persons 70 years and older (*p* trend < 0.0001). CNS manifestations were reported for all deaths except one case with missing information. 

Complete information on vaccination status was available for 5678 (93.7%) of all cases. Of these, 75 (1.3%) were adequately vaccinated. A further 314 (5.5%) were inadequately vaccinated, of whom 122 had received only one TBE vaccine dose, and the remainder two or more, but either too recently (<3 weeks) or too long before illness onset (> 1/> 3/> 5 years, depending on dose number and age) to provide protection (for details, see methods). The proportion of cases with adequate vaccination status was slightly higher among patients under the age of 15 (2.0%) and those 70 years or older (1.8%). 

## 3. Discussion

Our overview shows that TBE has become endemic in the majority of southern German districts, with slower contiguous north-eastern and more patchy north-western spread. The observed seasonal variation and marked variability in regional and annual TBE incidence in Germany is characteristic of this disease, as evidenced by similar patterns reported from other European and European Free Trade Association (EFTA) countries [35,36]. Comparison of absolute TBE incidence in Germany with that in other countries is difficult due to differences in surveillance systems, case definitions, vaccination coverage as well as marked regional differences in TBE incidence within countries [35,37,38,39]. For instance, in 2012, 9 of 20 countries included only cases with neurological manifestations in their TBE case definition, while others, like Germany, include cases with non-specific symptoms as well [35]. Nonetheless, as summarized in a recent European Centre for Disease Prevention and Control (ECDC) publication, annual TBE incidences reported from 2012 to 2016 were markedly higher in the Baltic States (9.5–15.6 cases/100,000 inhabitants), Slovenia (7.0) and the Czech Republic (4.8), followed by Sweden and Slovakia (both 2.4), with other countries reporting TBE incidences under 1 case/100,000 inhabitants, i.e. comparable to or lower than those in southern Germany [39]. 

Reasons for the observed annual variations in incidence, including the marked increase in the most recent seasons 2017 and 2018, are likely multifactorial and interdependent, related to climatic conditions, variations in both tick as well as host populations, TBEV prevalence in ticks and human behavior. Within endemic areas, however, social and behavioural factors that increase human exposure to ticks appear to be the most important determinants of TBE incidence [40,41,42]. For instance, Randolph et al. [43] showed similar tick abundance in regions with very high TBE incidence in 2006, but much lower incidence in 2007, despite differing weather conditions in the two years. Thus the most likely explanation put forward for the incidence spike in 2006 was weather-related human activity, such as mushroom foraging as described in the Czech Republic in 2006 [44]. 

The demographic distribution of notified TBE cases in Germany is similar to other reports of higher TBE risk in older adults and in males of all age groups [1,45]. The prevalence of antibodies against *Borrelia burgdorferi*, the cause of Lyme Disease—likewise transmitted mainly by *I. ricinus* in Germany—was also higher in males (5.5%) than females (4.1%) in Germany from 2003–2006 [46]. This suggests more frequent tick exposure on the part of boys and men, perhaps related to specific outdoor activities or occupations. On the other hand, in a seroprevalence study performed from 1980–1985 using a TBE-specific neutralization test on residual sera from clinical laboratories obtained from 8526 patients in TBE endemic regions [19], seropositivity increased with age, but there was no difference between men and women. This would suggest that TBE infection in males is more likely to lead to symptomatic, and thus ascertainable disease. In fact, we did observe a more severe disease course in male vs. female cases in our data. Sex-specific seroprevalence was unfortunately not reported in a number of similar more representative, albeit smaller, recent studies performed in Germany [47] or Scandinavia [48,49]. Thus further seroprevalence studies with test systems differentiating between infection-induced versus vaccine-induced antibodies would be useful. 

From 2001 to 2018, we found a slight—2% annually—but significantly increasing trend in TBE incidence. This was primarily due to the high case numbers in 2017 and 2018, as evidenced by the absence of an increasing trend from 2001–2016. Due to the fluctuating nature of TBE incidence, only future seasons will show whether this may be the beginning of a sustained upward trend. 

From 2007 to 2018, the number of districts classified as TBE risk areas increased from 129 to 161. Most newer risk areas bordered on existent ones in BW, BY, Hesse and Thuringia. The most notable recent developments were first-time risk areas in Saarland (2010), Saxony (2016) and Lower Saxony (2019). Mean annual incidences from 2014 to 2018 varied widely among risk areas, ranging from 0 to 9.6 cases/100,000 inhabitants. A sustained decrease in TBE incidence in recent years was observed in a region consisting of 5 districts in norther BW and BY, despite only low to moderate vaccination coverage. We recommend further investigation of local climatological or ecological changes to identify factors advantageous or disadvantageous to circulation of TBEV in new endemic areas or areas with sustained elimination, respectively.

Over half the sporadic autochthonous cases reported from non-risk areas in the past five years clustered around extant risk areas in Bavaria, and Hesse, Thuringia and Rhineland-Palatinate. However, the remainder occurred in Lower Saxony, where one district was classified as a risk area for the first time in 2019, in Northrhine-Westfalia and in the Eastern federal states of Brandenburg, Mecklenburg-Western Pomerania and Saxony-Anhalt (Figure 5b). In the latter, TBE was endemic in the 1960s [10,11]. In Lower Saxony, TBEV was detected in ticks in areas reported as the PoI [50] in several cases. In addition, serological TBE-antibody monitoring of forestry workers revealed asymptomatic seroconversions in 9 unvaccinated workers from 2006–2014 (total blood samples tested: 4609), although no clinical cases were detected [51]. TBE emergence with 2 TBE cases detected in 2016 was also recently described in the adjacent Netherlands, which borders on Lower Saxony and Northrhine-Westphalia [52,53,54]. 

Nonetheless, the majority of all cases occurred in southern Germany, even though the main TBEV vector *I. ricinus* is endemic throughout Germany. Reasons for this remain poorly understood. TBEV prevalence in *I. ricinus* in southern Germany is low, most often <1%, with higher prevalences observed in some small studies investigating engorged ticks (reviewed in [55]). This implies that sustained TBE circulation in tick foci is not easily established, in keeping with the rather slow geographic spread observed in Germany. In fact, because viraemia in animal hosts is only transient, transmission from viraemic animals to ticks is thought to play a minor role in sustaining TBE infection in ticks [56]. Rather, TBEV transfer from infected to uninfected ticks during co-feeding on the same animal host—most efficiently from infected nymphs to uninfected larvae—is thought to be a prerequisite for sustained TBEV transmission [57,58,59,60,61]. Randolph has provided evidence that TBEV persists only in regions where climatic conditions favour the simultaneous feeding of these two immature tick stages [61]. Although this may help explain the TBE distribution in Germany, no data on tick synchrony or analyses of climate differences between endemic and non-endemic regions are thus far available to provide definitive evidence. 

The primary goal of defining TBE risk areas in Germany was to provide an overview of districts in which preventive measures, primarily vaccination, should be implemented and to visualize TBE spread over time (Figure 5a,b). The advantages of our incidence-based approach to defining risk areas based on surveillance data since 2002 and taking into account TBE incidence in neighbouring districts include a relative stability of risk estimates in the face of small case numbers and possible uncertainty regarding the exact PoI across district borders. Nonetheless, our approach has some limitations. Most importantly, the use of administrative districts as the underlying geographic unit does not permit risk assessment at the level of smaller geographic units, which might better reflect the patchy nature of TBE foci. In addition, administrative reforms have led to amalgamation of smaller districts into larger ones in recent years. Furthermore, TBE incidence in bordering areas of neighbouring countries is not taken into account, in particular the relatively high incidence reported from Czech districts bordering on Saxony [12]. Finally, if vaccination coverage should increase further, human cases would become a poor correlate of infection risk. To some extent, these limitations are compensated for by the use of notification data dating back until 2002 as well as by the low incidence threshold used to classify districts as risk areas. 

In the long term, however, complementary TBE risk indicators would be valuable. Monitoring TBEV in ticks has occasionally been useful to confirm TBEV transmission in areas of low human incidence, for instance in Mecklenburg-Western Pomerania [62] and Lower Saxony [50,63]. However, as prevalence in ticks is usually <1% [55,64], large numbers of ticks must be collected to detect TBEV, making this an impractical approach for systematically monitoring large areas and explaining why attempts at isolating TBEV from ticks to confirm TBEV transmission have frequently been unsuccessful [55]. Instead, TBE seroprevalence in animal hosts with a defined spatial radius may be a more useful indicator, as reviewed in [55,64]. However, requirements for a broader application of such an approach include the validation and standardization of serological testing, excluding cross reactions with other flavivirus infections (e.g. West Nile virus) as well as characterization of the duration of antibody persistence following TBEV infection in different animals. In addition, correlation with human disease must be confirmed. Some of these aspects are being addressed in ongoing projects within the interdisciplinary national research network TBENAGER (Tick-borne encephalitis in Germany) [65].

Vaccination coverage in TBE risk areas, currently available only for children at school entry, was very heterogeneous and in most districts insufficient to lead to sustained reduction of TBE incidence, such as observed in Austria at a coverage of ~85% [66]. Moreover, data from previous years show that vaccination coverage in children has actually decreased since 2009/2010 in many risk areas [67]. In addition, surveys of a household-based health panel representative at the state level showed lower coverage than at school entry in the past [67]. In the future, we plan to estimate district-based vaccination coverage in older age groups based on health insurance claims data. Increasing vaccine uptake especially in districts with the highest incidences could achieve a marked reduction in TBE disease burden, as over 70% of all cases occurred in only one third of all risk areas. Only a small proportion of cases was reported as adequately vaccinated, thus the vast majority could have been prevented by vaccination.

Our analysis of available clinical data is in keeping with a more benign course of disease particularly in very young children and more severe manifestations in older adults. Almost one-third of adults 70 years of age and older were reported with the most serious manifestations of encephalitis or myelitis, in line with observations by Kaiser [3,45,68]. Case fatality was also highest in this age group. Although CNS symptoms were reportedly lacking in close to half of notified TBE patients, 79.4% of these were hospitalized. This suggests that reporting of more severe CNS manifestations may be incomplete. We are investigating this further in an on-going case-control study with detailed ascertainment of acute and long-term TBE manifestations in Germany within the TBENAGER framework [65] (see above). In addition, we are planning a detailed analysis of clinical manifestations in relation to vaccination status.

## 4. Materials and Methods 

### 4.1. TBE Surveillance in Germany

TBE became statutorily notifiable in Germany in 2001, requiring transmission of case-based data from local health authorities to the state and national levels according to a standardized case definition. Fulfilment of the TBE case definition requires (either non-specific symptoms (at least 2 of the following: chills, severe malaise, headache and muscle, limb or back pain) or signs of central nervous system (CNS) infection (meningitis, encephalitis or myelitis separately or in combination)) and [laboratory confirmation by means of either simultaneously elevated IgM and IgG TBE-specific antibodies in serum or cerebrospinal fluid (CSF) or an increase in TBE-specific IgG antibodies in serum or the detection of intrathecal antibody synthesis] [69]. However, until 2004, elevated IgM TBE-specific antibodies were sufficient for laboratory confirmation [70]. 

Diagnosis is generally performed in peripheral laboratories. Commercially available enzyme-linked immunosorbent assay (ELISA) test kits usually have a high validity in unvaccinated cases, but can be false positive test among cases with prior vaccination [71]. From 2016 to 2018, we matched statutory TBE surveillance data to results from confirmatory testing using immunofluorescence testing for antibodies to TBE and other flaviviruses that was performed for a small proportion of cases at the national TBE consultant laboratory. Of all notified cases, 5.3% (76/1417) were thus additionally confirmed. 

From 2001 to 2013 the following variables were reported regarding disease severity: presence or absence of non-specific symptoms, meningitis/encephalitis or myelitis; hospitalization and outcome (survived/died). From 2013 onwards meningitis and encephalitis were reported separately, although stable nationwide implementation was not achieved until 2015. Therefore, clinical manifestations were analyzed in detail for cases notified from 2015 to 2018 only. Information on vaccination status included the total number of doses received and the date of last vaccination. Vaccine doses were counted only if received at least 3 weeks prior to disease onset. Vaccination status was categorized as adequate if patients had received either 2 or 3 doses with the second or third dose less than 1 year or 3 years, respectively, prior to disease onset. When 4 or more doses were reported, vaccination was considered adequate if the last dose was received <5 and <3 years prior to disease onset in persons up to age 49 and those 50 years and older, respectively. The type of vaccine was available for only 49 (10.2%) of vaccinated cases; thus, we did not take into account the fact that persons vaccinated with FSME-Immun® (Pfizer, Berlin, Germany) require boosters every three years from the age of 60 rather than 50 years, as is recommended for Encepur® (Glaxo Smith Kline, Berlin, Germany).

The district of residence was reported as a possible place of infection (PoI) in 88.4% of cases with a reported PoI and Germany given as the only possible country of infection (4471/5060). Of these, 4933 cases were reported with a single PoI. When >1 PoI was reported or the PoI was missing, but Germany was reported as the only country of infection, we proceeded as follows for incidence calculations, (including Figure 1): When 2 or 3 districts were reported as possible PoIs (n = 129), we chose the state of residence if ≥1 of the PoIs was located there (n = 111) or the state of the PoI listed first (n = 18). When missing, the PoI was assumed to be the state of residence if the case resided in a risk area or in a federal state with at least 2 risk areas (537 additional cases). If a PoI in Germany was reported together with another possible PoI in a foreign country, the PoI was considered missing (n = 35). Note that for calculations pertaining to TBE risk areas (see below) only cases with reported PoI were included. 

We analysed data related to TBE cases notified from 1 January 2001 to 31 December 2018 as available at the Robert Koch Institute (RKI) on 17 January 2019 using STATA 15 (StataCorp LP, College Station, TX, USA). We used an age-adjusted negative binomial regression model to estimate the temporal TBE incidence trend from 2001 to 2018, calculating incidence rate ratios (IRR) and 95% confidence intervals (CI). We compared proportions using the chi-squared test. *P*-values < 0.05 were considered statistically significant. 

### 4.2. Definition of Risk Areas

An incidence-based definition for TBE risk areas based on statutory surveillance data was developed at RKI in 2006 in a consensus process involving TBE experts including microbiologists, clinicians and epidemiologists for implementation in 2007 [34]. Administrative districts were retained as the underlying geographic unit for risk area definition due to lack of more detailed geographic data regarding PoI. In addition, TBE cases occurring in the area surrounding each district were incorporated into the new definition to account for mobility of exposed persons as well as to achieve smoothing between neighbouring districts. To accomplish this, TBE incidence was calculated in a second geographic unit, the ‘district region’, consisting of the respective district plus all adjoining districts. Differences in the variance of TBE incidence resulting from differences in underlying population size in the respective districts/district regions were accounted for by applying a probabilistic approach [72]. Thus, a district was defined as a risk area if the incidence in the district or respective district region was significantly above the chosen incidence threshold of 1 TBE case/100,000 inhabitants in any 5-year period since 2002, meaning that the number of observed cases was significantly greater (*p* < 0.05 according to the Poisson distribution) than the number of cases required to reach the threshold incidence given the population size (expected number). When 2 or 3 possible PoI were reported, 0.5 or 0.33 cases were assigned to the reported districts, respectively. If more than 3 PoI were reported, the PoI was considered unknown.

In choosing an incidence cut-off, there was consensus that the majority of risk areas defined prior to 2007 should be captured by the new approach. As it would be difficult to communicate the disappearance of a perceived public health risk based solely on a change in definition, it was decided to retain the risk status for any districts that might not be defined as risk areas with the new approach. Furthermore, although incidence was considered a better measure of risk than absolute case numbers, it was recognized that the population denominator would not accurately reflect the number of persons at risk because some persons lack a risk for exposure and some are vaccinated. Thus, calculated incidence would underestimate true risk as vaccination coverage increased, justifying a lower rather than higher incidence cut-off. Taking these issues into account, an incidence cut-off of 1 TBE case/100,000 population/5 years was chosen, close to the nation-wide five-year incidence of 1.3 in 2002–2006. In a further expert consultation in 2011, it was decided that districts classified as TBE risk areas would remain so for a period of at least 20 years.

### 4.3. Vaccination Coverage

TBE vaccination status (total doses received) is ascertained at school entry in 4 to 7-year-old children by the district health department, which reports aggregate data to RKI via the state health department according to the IfSG. Approximate vaccination coverage in risk areas was calculated as the percentage of children who had received ≥3 doses of a TBE vaccine based on data received 2017 from the four states with the majority of TBE risk areas (BW, BY, Hesse and Thuringia, n = 251,825 children with vaccination records; 92.5% of all children examined). 

## 5. Conclusions

TBE is endemic primarily in southern Germany, with a more recent contiguous north-eastward spread into Saxony and more patchy spread to north-western regions. Within endemic regions, there is marked temporal and geographical variation of TBE incidence. Despite some limitations, the definition of risk areas has been helpful for formulating vaccination recommendations and monitoring the spread of TBE in Germany. Increasing vaccination coverage especially in high incidence areas would markedly decrease the number of infections. 

## Figures and Tables

**Figure 1 pathogens-08-00042-f001:**
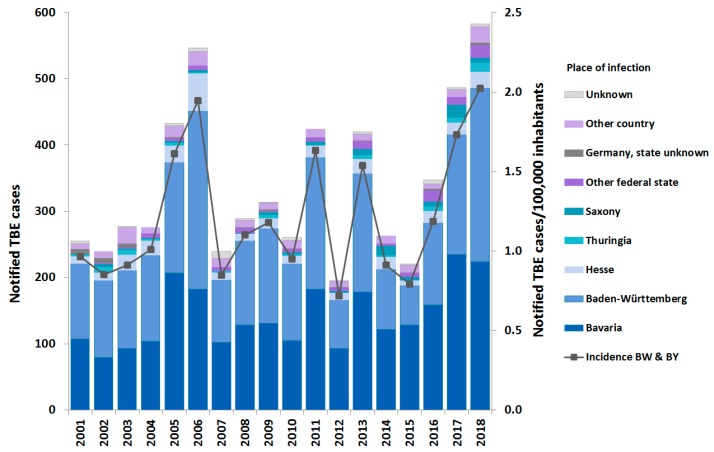
Notified tick-borne encephalitis (TBE) cases (N = 6,063) according to state of infection (left Y-axis) and incidence in Baden-Wurttemberg and Bavaria, Germany, 2001–2018 (right Y-axis).

**Figure 2 pathogens-08-00042-f002:**
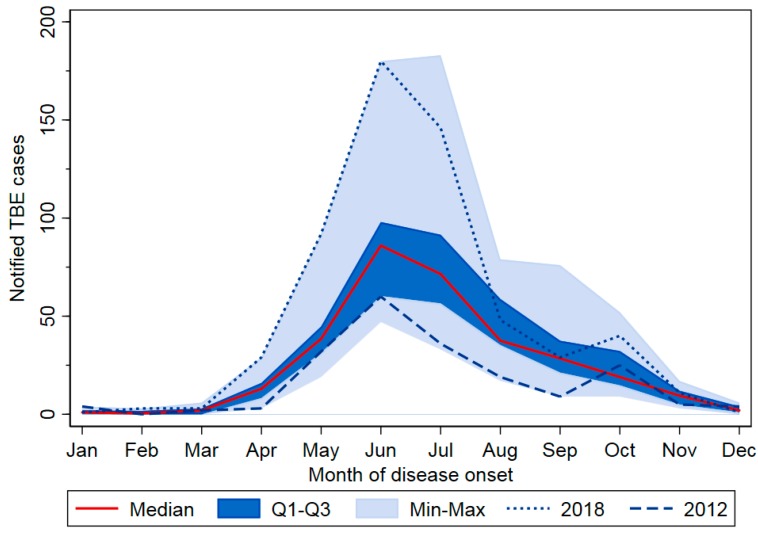
Notified TBE cases (N = 6,063) according to month and year of disease onset, Germany, 2001–2018. Seasons with lowest and highest case numbers are highlighted (2012: 195 cases; 2018: 583 cases). Q1–Q3 = quartile 1–3, Min–Max = minimum–maximum.

**Figure 3 pathogens-08-00042-f003:**
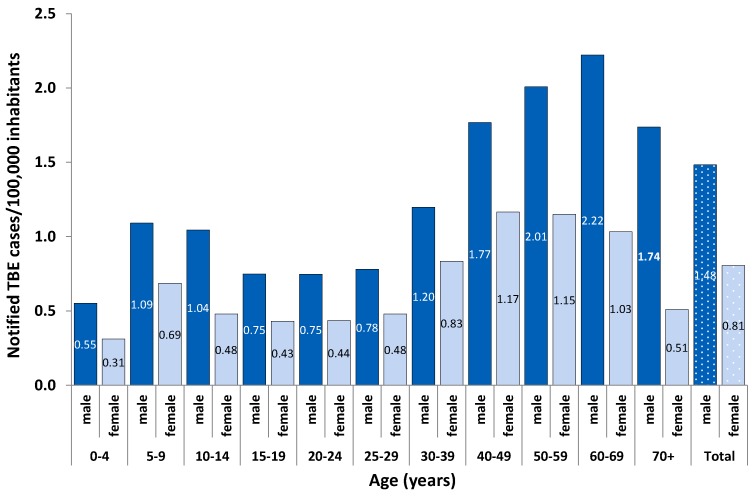
Mean annual incidence of notified TBE cases per 100,000 inhabitants according to sex (blue: dark male; light blue: female), Baden-Wurttemberg and Bavaria, Germany, 2001–2018 (n = 4977).

**Figure 4 pathogens-08-00042-f004:**
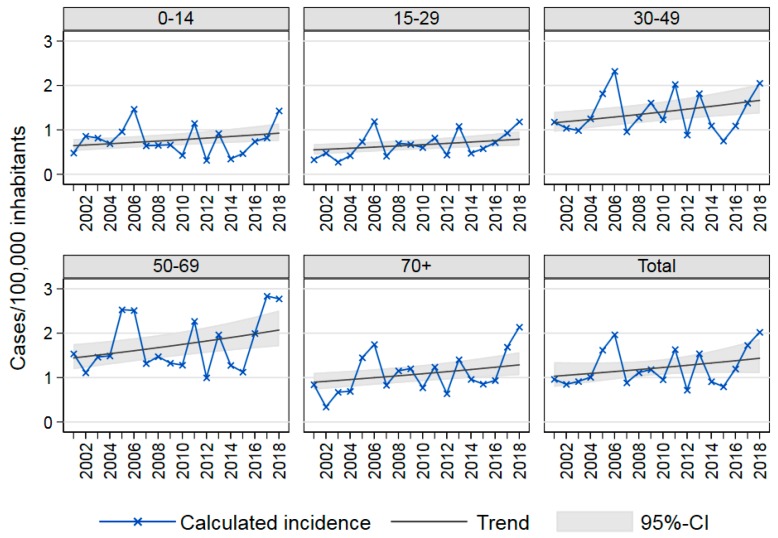
Age-specific TBE incidence Baden-Wurttemberg and Bavaria, Germany, 2001–2018, and trends as estimated in an age-adjusted negative binomial regression model: Total incidence rate ratio (IRR): 1.02, 95% confidence interval (CI): 1.01–1.04.

**Figure 5 pathogens-08-00042-f005:**
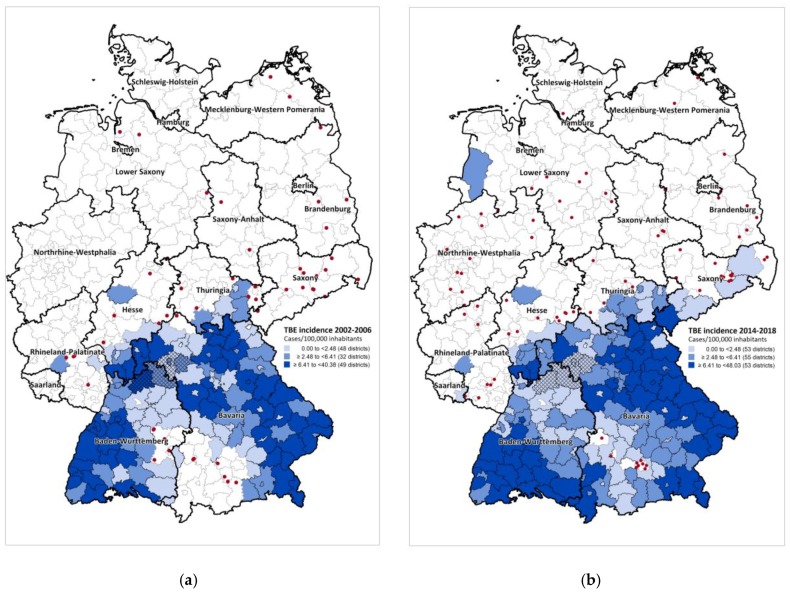
Districts defined as risk areas, Germany: (**a**) 2007 (n = 129, based on 1430 cases notified in 2002–2006); (**b)** 2019 (n = 161, based on 5,090 cases notified in 2002–2018). In 2007, 80 districts were defined as risk areas based on an elevated incidence in the district itself, 47 based on an elevated incidence in the ‘district region’ only and 2 based on past definition (for details, see methods). In 2019, 161 districts were defined as risk areas, 130 based on an elevated incidence in the district itself and 34 based on an elevated incidence in the ‘district region’ only and 2 based on past definition (see methods). Five districts in Rhineland-Palatinate were not classified as risk areas despite an elevated incidence in the associated district region (see methods), since no case was ever reported there and they were separated from bordering TBE-endemic districts by the Rhine River, considered a plausible natural boundary. Shading depicts districts classified as risk areas categorized according to TBE incidence 2002–2006 (a) and 2014–2018 (b). Dots depict sporadic cases in non-risk areas in the same respective periods ((a): n = 51, (b): n = 83 cases reported with a single known place of infection (PoI)). With the exception of one possible case (see text), no cases were reported in the 5 hatched districts (LK Neckar-Odenwald-Kreis, LK Main-Tauber-Kreis, SK Würzburg/LK Würzburg, and LK Kitzingen) from 2014–2018.

**Figure 6 pathogens-08-00042-f006:**
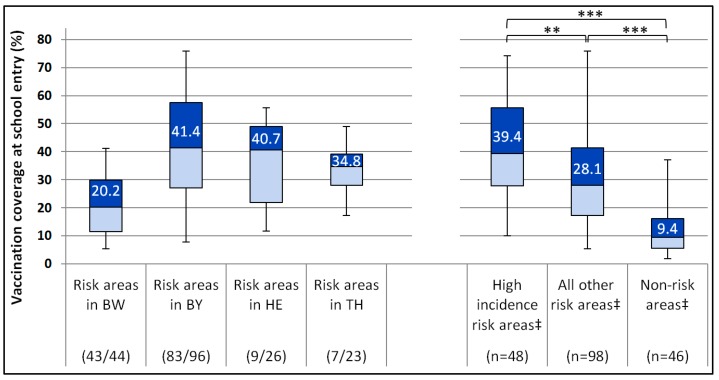
TBE vaccination coverage (VC, ≥3 doses) by federal state (Baden-Wurttemberg (BW), Bavaria (BY), Hesse (HE) and Thuringia (TH)) and TBE incidence category (high incidence: >6.4 TBE cases/100,000 inhabitants). VC in BY and HE ascertained in 2016, in BW & TH in 2017. Numbers in boxes = median VC. ** *p *< 0.007 and *** *p *= 0.0001, Kruskal–Wallis test. ^‡^In BW, BY, HE and TH. Fractions in brackets under x-axis: number of risk area districts/total number of districts in each state in 2017.

**Figure 7 pathogens-08-00042-f007:**
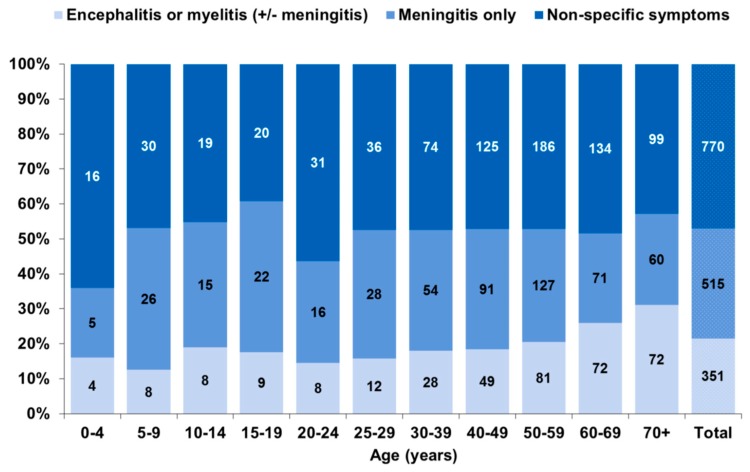
Notified TBE cases according to clinical manifestations, Germany, 2015–2018 (n = 1636).

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
