# Peer review of "Epidemiology of Tick-Borne Encephalitis (TBE) in Germany, 2001–2018"

_pathogens, 2019, doi:10.3390/pathogens8020042_

Round 1
Reviewer 1 Report
The manuscript by Hellenbrand et al., describes the current status of TBE in Germany and compares the current state with the changes in the epidemiological picture over the past 17 years. In general, the paper shows that many of the foci that existed in 2001 still exist, and that the TBE virus appears to be spreading into new areas in southern Germany. In addition, in some areas where the virus was known to exist, the case load is increasing. Most other risk factors for disease , including age and vaccination status, appear to be consistent over time as does the disease clinical presentation. This study again emphasizes the importance of childhood vaccinations and maintenance of vaccination status for those living in areas endemic for TBE virus. This paper is very well written and comprehensive it its presentation regarding TBE in Germany.
Minor comments:
Lines 113-114: It isn’t clear to me why the authors opted to exclude data from 2017-2018 from their preliminary analyses. I think theses data should be included as 1. They may be the new normal and 2. They don’t seem markedly different from the peak in cases in the 2005-2006 time frame. Were there perhaps environmental factors (e.g. warmer winter) that might account for an increase in the tick population and hence, human TBEV infection?
Line 207: What is meant by “too long” or “too recent” since vaccination? Can you put a number to this? Such as less than 10 days or more than 20 years?
Line 209: These data are consistent with a paper that just came out from Hansson et al (PMID: 30843030) in a retrospective study of vaccine failures in Sweden. Consider citing.
Author Response
Please see point-by-point responses in the attached file.

Reviewer 2 Report
The authors have summarized the data needed to understand the epidemiology of tick-borne encephalitis in Germany that is needed to understand the need for vaccination. I could understand the role of reporting in comparing data over time and between countries. I can see that the data suggest the need to understand the role of the tick versus the behaviour of the host.
I appreciate the patient clinical information. Only 75 (1.3%) of the patients were adequately vaccinated according to the authors.
The paper will allow policymakers the information they need to adopt policy for different regions in Germany.
Author Response

(The authors gave the same response as above.)
